# Enhancing the communication of radiation exposure data for radiological workers in Korea using data visualization techniques

Kyoungho Choi

Department of Radiological Science, Research Institute of Health Statistics, Jeonju University, Jeonju, Republic of Korea

* ckh414@jj.ac.kr

**Data availability statement:** All relevant data are within the manuscript.

**Funding:** The author(s) received no specific funding for this work.

**Competing interests:** The authors have declared that no competing interests exist.

## Abstract

Effective communication of radiation exposure data is essential for improving safety management practices for radiological workers. However, traditional tabular formats used in reporting radiation exposure data often fail to convey critical patterns and trends, making it difficult for non-experts to interpret and act on the information. This study evaluates the application of data visualization techniques, including radar charts, box plots, sparklines, and Chernoff faces, to enhance the accessibility and comprehension of radiation exposure data. Using datasets from the "2022 Annual Report on Individual Exposure Doses of Radiological Workers" published by the KDCA, this study demonstrates how visualization can effectively highlight disparities across professions, demographic groups, and geographic regions. The findings underscore the significant potential of visualization methods in simplifying complex datasets, enabling stakeholders to make more informed decisions. Nonetheless, the study has limitations, including its reliance on pre-existing public datasets and a lack of real-time or granular data. Future research should focus on collecting primary data to explore causal relationships in radiation exposure trends and on applying advanced statistical and machine learning techniques to uncover deeper insights. By integrating robust visualization methods, this study aims to bridge the gap between raw data and actionable knowledge, ultimately contributing to safer occupational environments for radiological workers.

## Introduction

The clinical use of radiation has become indispensable in modern medicine, significantly contributing to advancements in diagnostic and therapeutic procedures [1]. However, the widespread use of radiation in healthcare settings has led to increased occupational exposure for medical professionals, particularly radiological technologists (RTs). This exposure poses serious health risks, including stochastic effects

such as cancer, which necessitate rigorous safety measures and effective monitoring systems [2,3]. The long-term health and safety of these workers depend on reliable and accessible communication of radiation exposure data, enabling informed decision-making and improved safety management practices.

To address these risks, the Korean Disease Control and Prevention Agency (KDCA) publishes an annual report on the individual radiation exposure doses of healthcare professionals. While comprehensive, these reports are predominantly presented in tabular formats, which, despite their accuracy, fail to effectively convey key trends, patterns, and disparities to non-expert audiences [4]. This limitation hinders the ability of stakeholders, including radiological workers and policymakers, to fully utilize the data for actionable insights. The complexity of radiation exposure data and its implications demand alternative approaches that enhance its interpretability and accessibility.

Data visualization techniques have been widely recognized for their potential to simplify complex datasets and reveal underlying pattern [5]. By transforming raw data into intuitive visual formats, visualization methods enable users to quickly grasp critical information, facilitating better communication and decision-making. For instance, radar charts and box plots can illustrate variations in radiation doses across different professions and demographic groups, while sparklines and Chernoff faces can reveal temporal trends and multivariate relationships [6,7]. Despite these advantages, the application of data visualization techniques to radiation exposure data remains underexplored, particularly in the context of occupational health.

This study aimed to assess the effectiveness of data visualization techniques in enhancing the communication and understanding of radiation exposure data. By applying methods such as radar charts, box plots, sparklines, and Chernoff faces to the KDCA's 2022 Annual Report on Individual Exposure Doses of Radiological Workers, this study aims to identify key trends, disparities, and patterns in radiation exposure. The study also evaluates the potential of visualization methods to empower stakeholders, including radiological workers and policymakers, by providing actionable insights that support safer occupational environments.

Furthermore, this study addresses the limitations of existing data communication practices and highlights opportunities for future research. Current reporting relies heavily on static, tabular datasets, which lack real-time and region-specific details essential for comprehensive analysis. Future research should focus on integrating real-time data collection systems and advanced statistical techniques to explore causal relationships and emerging trends in radiation exposure [8,9]. By bridging the gap between raw data and user-friendly information, this study contributes to the broader effort to improve safety management practices and promote evidence-based policy decisions in occupational health.

## Materials and methods

### Data source

For the analysis, relevant datasets were extracted from the data presented in the 2022 Annual Report, an official annual report published by the KDCA. This study processed and utilized data on the annual distribution of exposure doses among

radiological workers categorized by occupation, age, and sex, as provided in tabular format in the report. The report noted that safety management for diagnostic radiological workers in South Korea was first implemented under the Regulations on the Safety Management of Diagnostic Radiological Equipment. The number of diagnostic radiological workers was 12,685 in 1996, during the early stages of implementation, and increased by 8.4 times to 106,165 by 2022. This increase can be attributed to the rapid increase in the number of diagnostic radiological examinations, such as health screenings, propelled by improved medical infrastructure and heightened public interest in health. Although individual exposure doses received by radiological workers have also been increasing, due to safety management efforts, the average dose in 2022 was 0.38 mSv, similar to that in 2021. This value represents a significant decrease compared to the value in 2004 (0.97 mSv) but remains higher than those in developed countries such as the United Kingdom (0.066 mSv), Germany (0.31 mSv), and France (0.27 mSv).

## Methods for visualizing statistical information

Visualizing healthcare data generated from various forms and sources helps to uncover relationships and patterns, making it easier for readers to understand the data [10]. Controversies remain regarding the most suitable forms of graphic data presentation. An excellent statistical graphic portrays the maximum amount of information in the shortest amount of time, using the least amount of ink and the smallest space [6]. This section explores different methods for visualizing statistical information derived from healthcare data.

### Radar chart

A radar chart is a valuable tool for comparing the characteristics of objects, people, or groups based on various evaluation factors or criteria. This tool can become less effective when there are too many items, as interpretation can become complex. However, radar charts provide highly useful statistical information for comparisons when there are approximately five items. By comparing Table 1 with Fig 1, it is evident that product comparisons are more intuitively understood when they are expressed in a radar chart than when they are expressed in a table. The chart shows that Product A is robust, cost-effective, and convenient to use, whereas Product B is superior to Product A in terms of creativity and design.

### Sparkline chart

As illustrated in Fig 2, sparklines are miniature line charts that are typically drawn without axes or coordinates. They are particularly effective in visualizing trends in continuous measurement variations, such as temperature changes or stock market price movements, in a simple and compact format. When created using Excel, Sparklines are embedded within individual cells, making them particularly useful in scenarios where space is limited or when the primary objective is to highlight qualitative changes rather than precise numerical values. This format enables users to view the data and graphical representations simultaneously.

**Table 1. Data organized in tabular format.**

|  | Product A | Product B |
|---|---|---|
| Robustness | 32 | 12 |
| Cost | 32 | 12 |
| Convenience | 28 | 12 |
| Design | 13 | 35 |
| Creativity | 16 | 28 |

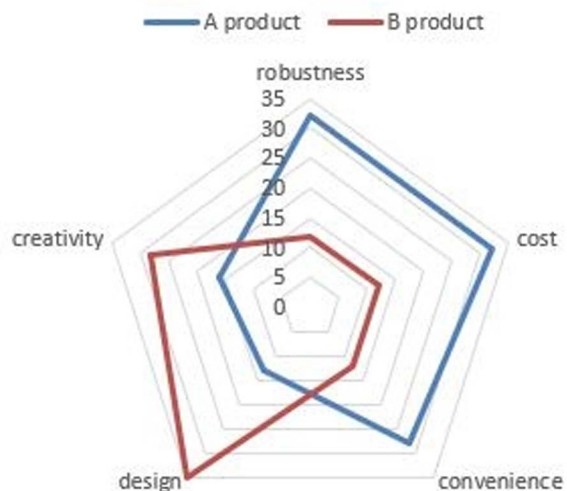

**Fig 1. A radar chart is a useful diagram for comparing the characteristics of objects, individuals, or groups based on various evaluation factors or criteria.**

| | A | B | C | D | E | F |
|---|---|---|---|---|---|---|
| 1 | | x value | y value | z value | Sparkline1 | Sparkline2 |
| 2 | KOR | 175.1 | 40.0 | 90.3 | | |
| 3 | KOR | 10.0 | 20.0 | 30.0 | | |
| 4 | KOR | 70.9 | 174.1 | 103.2 | | |
| 5 | KOR | 46.2 | 166.7 | -20.0 | | |
| 6 | KOR | 0.0 | -100.0 | 94.9 | | |
| 7 | KOR | 76.7 | 168.4 | 91.7 | | |
| 8 | USA | 43.9 | 156.6 | 112.7 | | |
| 9 | USA | 261.9 | 97.7 | 162.1 | | |
| 10 | USA | 159.2 | 184.6 | 245.1 | | |
| 11 | USA | 146.8 | 159.6 | 200.5 | | |
| 12 | USA | 159.4 | 42.3 | 117.1 | | |

**Fig 2. Sparkline chart illustrating annual trends in radiation exposure doses over time.** This compact visualization highlights overall trends without detailed axis labels.

## Sunburst chart

The sunburst chart is named after its resemblance to the radiating rays of the sun. Although it is similar to a donut chart, it is better suited for representing hierarchical data, such as the data in Table 2. In the sunburst chart, each concentric ring represents a different hierarchical level, with the innermost circle representing the highest level. As shown in Fig 3, sunburst charts are particularly effective for comparing and examining the characteristics of each level in the hierarchy, using distinct colors to differentiate between categories. This visualization technique is widely used in healthcare data analysis, particularly for data with regional hierarchical structures.

## Map chart

Map charts are valuable tools for examining regional characteristics by visualizing data on maps. Fig 4 shows a map chart illustrating murder and non-negligent manslaughter crime rates in the United States in 2012 [11]. To create a map chart

**Table 2. Hierarchical structure in tabular format.**

| | | | Value |
|---|---|---|---|
| State 1 | | | 9 |
| | Region 1 | | 25 |
| | Region 1 | City 3 | 23 |
| | Region 2 | | 24 |
| | | City 5 | 89 |
| | | City 6 | 22 |
| | | City 7 | 12 |
| State 2 | | City 8 | 18 |
| | | City 9 | 87 |
| | Region 4 | City 10 | 88 |
| | Region 4 | City 11 | 17 |
| State 3 | | City 12 | 9 |
| | Region 5 | | 25 |
| | Region 6 | City 14 | 24 |
| | Region 6 | City 15 | 89 |
| | Region 6 | City 16 | 16 |

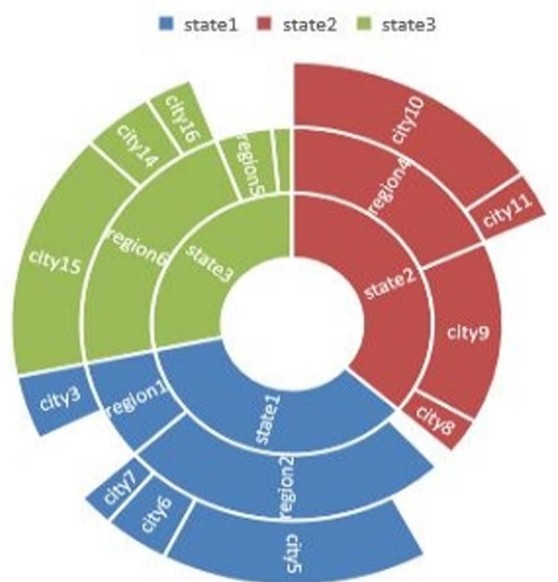

**Fig 3. A sunburst chart is a useful visualization tool for healthcare data with a regional hierarchical structure.** By using different colors to represent multiple hierarchical levels, it effectively allows for comparisons of the characteristics of each level.

in Excel, users can utilize the 3D map features available from Excel 2013 and newer versions. In map charts generated using Excel, data values are typically represented as bar charts superimposed on the map. Alternatively, a heat map format in which higher values are shown as larger and redder areas, as shown in Fig 4, could also be effective. Although map charts are excellent for providing general regional comparisons, they are limited in terms of conveying precise quantitative information.

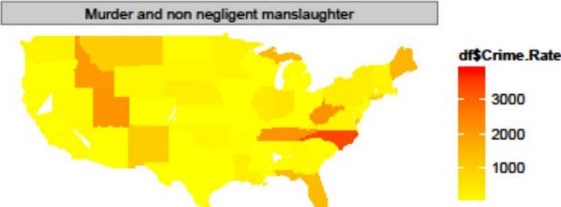

**Fig 4. U.S. state heat map illustrating murder and non-negligent manslaughter crime rates in 2012.** Reder regions indicate higher crime rates. Data source: Thoplan (2014).

### Chernoff faces

Chernoff faces, introduced by Chernoff in 1973, are a multidimensional data analysis technique that visually represents multivariate statistical data using images of human faces. When creating Chernoff faces on a two-dimensional plane, various facial features such as the width and height of the face, eyes, nose, mouth, and ears are substituted with different variables, enabling a clear understanding of data characteristics represented by up to 15 variables [12]. Chernoff faces are widely used to visualize the results of big data analyses in various fields. Notable Korean studies that used Chernoff faces include those on the representation of urban statistics through big data analysis [12], analysis of public library operation and usage by each local government [13], comparisons of different ranking systems in sports [14], and those involving big data statistical analysis of regional health-related data from the "Youth Life Survey" [15]. Other significant studies include one that visualized violence and property crime patterns by state in the United States [11] and a study that demonstrated five organic compounds in drinking water sampled from 12 regions in Poland, where features such as mouth curvature, nose size, and eyebrow direction and length corresponded to different variables [16]. Fig 5 shows

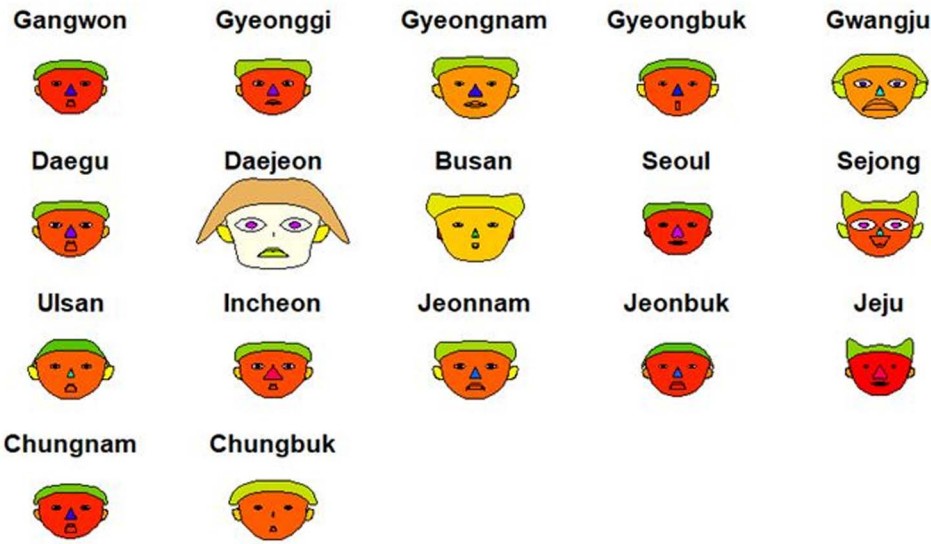

**Fig 5. Chernoff faces visualization of 17 metropolitan local governments in Korea.** Each facial feature is generated based on multivariate variables.

a representation of data using the Chernoff face method. In this figure, health statuses are categorized and compared by region using multivariate statistical models [15]. This chart was created using the R programming language, with face colors used for group categorization.

## Analysis tools

The analysis tools used to visualize statistical information in this study include Microsoft Excel, the Orange3 (https://orangedatamining.com/) program for big data analysis and machine learning accessible to beginners, and the R (https://www.r-project.org/) programming language for creating Chernoff faces.

## Results

### Descriptive statistics

Visualization is valuable in the early stages of data analysis to quickly grasp summary information. For example, a box plot provides insights into the central tendency and dispersion, making it particularly effective for intuitive comparisons between groups.

The 2022 Annual Report primarily presents information on average exposure doses by occupation in a tabular format, in addition to other graphical representations such as box plots or violin plots (Fig 6). Fig 6 not only displays the average and standard deviation for each occupation but also highlights that the exposure dose for RTs is significantly higher than that for dental doctors (DD), nurses (N), and general doctors (GD).

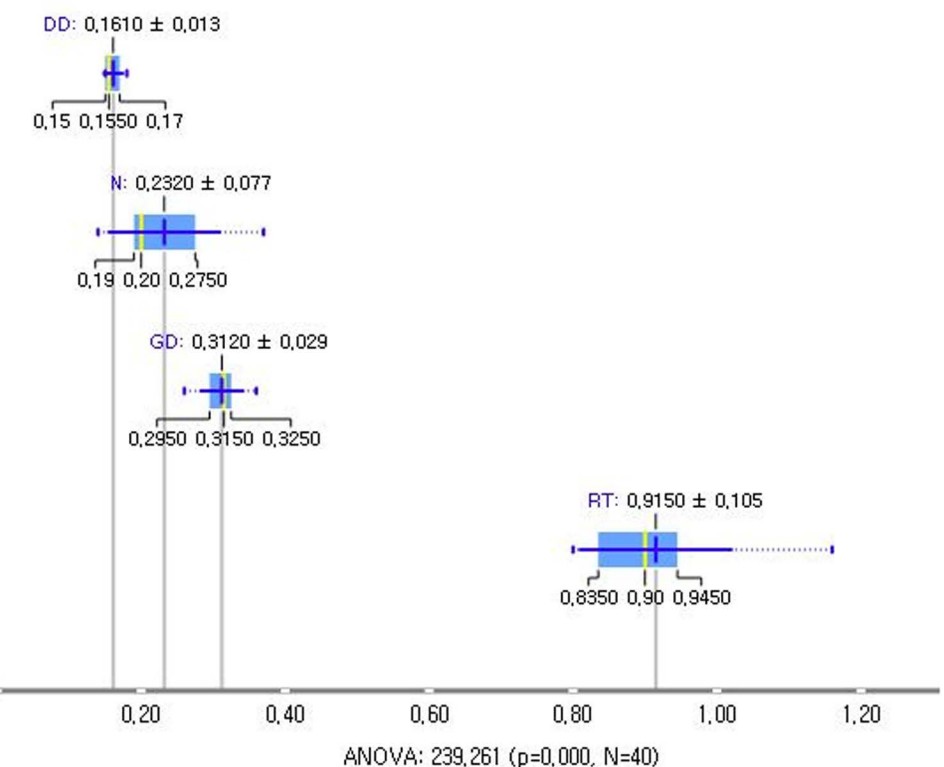

**Fig 6. Comparative box plots and violin plots of radiation dose distribution by occupation.** Significance testing of the mean can also be conducted using analysis of variance (ANOVA).

## Dose distribution by sex and age for radiological technologists

The average exposure dose for RTs was significantly higher than that for other occupations, necessitating an in-depth analysis of this group. A closer examination of the dose distribution by sex and age provides detailed information; however, the data presented in tabular format in the annual report makes comparisons difficult. Moreover, the complexity of the tables, with each cell containing count, percentage, and dose information, further complicates comprehension. To enhance understanding, a graphical representation of the dose distribution by sex and age, similar to that shown in Fig 7, would be highly beneficial. Fig 7 demonstrates two key findings: (1) RTs under the age of 25 years receive higher doses and (2) male RTs consistently experience higher exposure doses than their female counterparts across all age groups.

## Annual trends in the number of radiological workers by professional category

The 2022 Annual Report provides data on radiological workers reported to the Exposure Dose Management Center, totaling 106,165 workers as of 2022. RTs, GDs, and DDs comprised the majority of the workforce, accounting for 72% of the total. The number of workers in all categories increased, with the most significant growth observed among nurses (N) since 2018, followed by radiology residents (R). Despite this overall increase, identifying specific growth patterns solely from the quantitative data in the table can be challenging. Including a sparkline chart (Fig 8) would convey trends and growth patterns more intuitively, helping readers gain a clearer and more accurate understanding.

## Dose distribution by region and professional category

A remap chart was generated to compare the dose distribution of RT by region and professional category in 2022 (Fig 9). This chart represents dose sizes proportional to the area of each professional category. When created in Excel, the layout is dynamically adjusted as the chart is resized. Fig 9 shows that the average exposure dose for the RT group in the Jeonbuk region was higher than that in the Seoul and Busan regions. In Seoul and Busan, the average dose in the GD group was higher than that in the DD group. A map chart can be displayed on a map, enabling regional comparisons. The taller the bar, the greater the collective dose in that region. Presenting such map charts alongside tables in the annual report could significantly enhance readers' understanding.

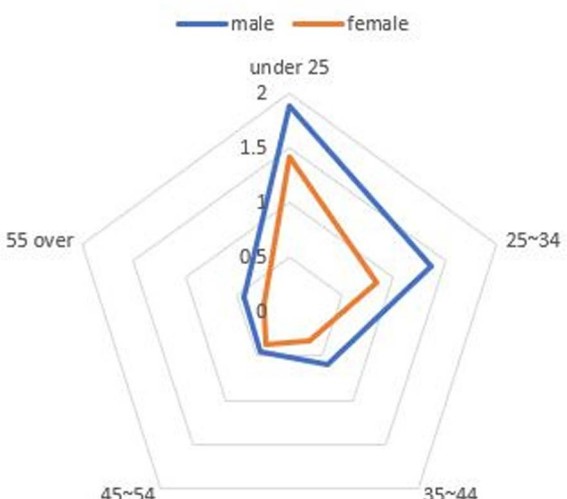

**Fig 7. Radar chart comparing radiation exposure doses by sex and age group.** The figure highlights differences in exposure levels across demographic categories.

| categoty | 2018 | 2019 | 2020 | 2021 | 2022 | | |
|---|---|---|---|---|---|---|---|
| RT | 26991 | 28476 | 29668 | 30945 | 32626 | | |
| GD | 19539 | 20539 | 21667 | 22951 | 24077 | | |
| DD | 18248 | 18950 | 19210 | 19720 | 20083 | | |
| R | 1835 | 2271 | 2315 | 2306 | 2363 | | |
| N | 8374 | 9382 | 10075 | 10664 | 11265 | | |
| NA | 1594 | 1807 | 1841 | 1951 | 2086 | | |

**Fig 8. Sparkline charts visualizing annual trends in the number of radiological workers by professional category.** The trend lines indicate variations in workforce distribution over time.

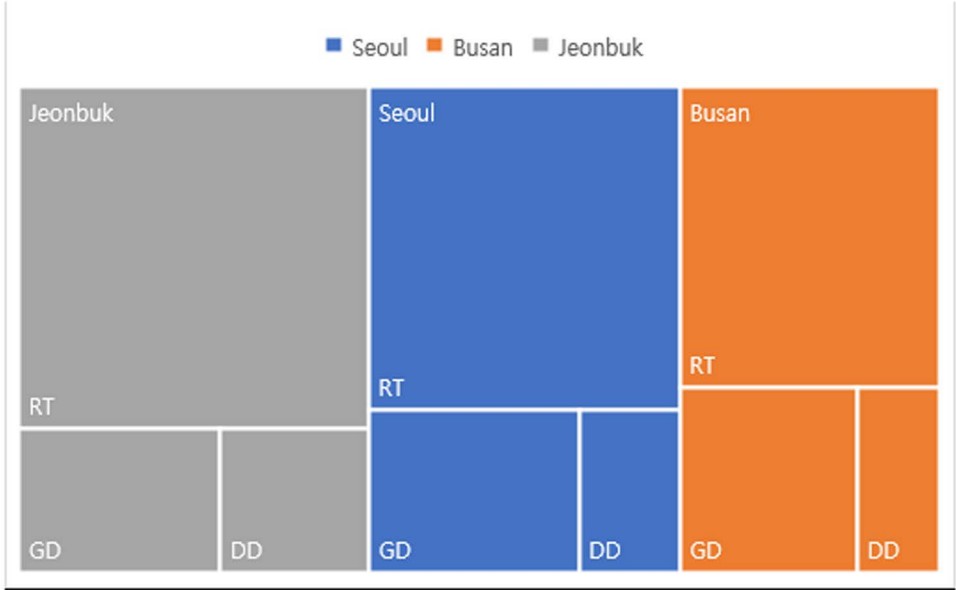

**Fig 9. Treemap chart illustrating radiation exposure doses across three professional categories in different regions.** The size of each section is proportional to the total dose received.

## Graphs using multivariate data

Fig 10 shows Chernoff faces that depict the dose distribution across the professional categories (RT, GD, DD, and N) using multivariate data. This chart was created for six major regions in Korea using the 'aplpack' package in R, which allows for the representation of different facial colors to convey data. Regions such as Incheon, Daejeon, and Jeonbuk, which are small but have red-colored faces, were identified as high-dose regions. Including this visual representation alongside tables in annual reports would significantly enhance our understanding of the exposure dose distribution across different regions.

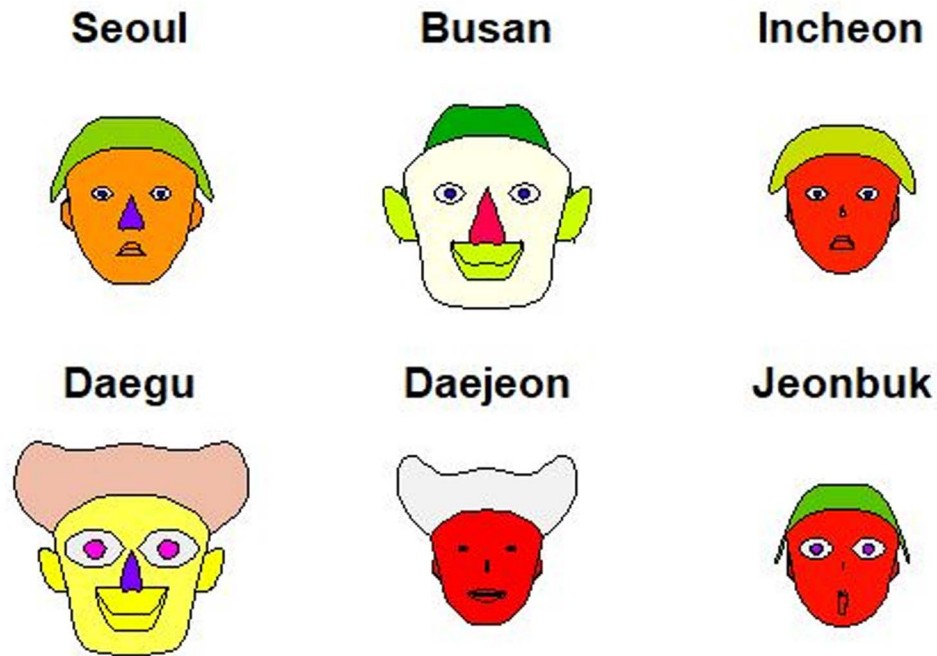

**Fig 10. Chernoff faces visualization for six regions in Korea, depicting variations in radiation exposure using multivariate facial representations.** The red-colored faces indicate regions with higher average doses.

## Understanding statistical trends

In Korea, due to the continuous strengthening of safety management, the average exposure dose for radiation workers has been decreasing since 2018, with the annual dose in 2022 remaining at 0.38 mSv, the same as in 2021. Despite this overall downward trend, exposure doses in the medical sector remain higher than those in other industrial sectors, such as nuclear power [17]. Furthermore, within the healthcare sector, RTs exhibited a significantly higher average exposure dose. Therefore, additional efforts to reduce the exposure are necessary. Fig 11 shows the estimated regression line for the average dose received by RTs over the past 10 years. Despite variations, a clear decreasing trend is observed. If this reduction is due to stringent safety management and regulatory measures, it can be inferred that these efforts are beneficial.

## Discussion

This study highlights the significant role of data visualization techniques in enhancing the accessibility and comprehension of radiation exposure data for radiological workers. Traditional tabular formats, such as those used in the KDCA's Annual Report, often present challenges for non-expert audiences, as these formats obscure critical trends, disparities, and patterns [4,6]. By transforming raw data into intuitive visual representations, this study demonstrates how visualization methods can bridge the gap between complex datasets and actionable insights [5,7].

The findings reveal several critical insights into radiation exposure trends. For example, radar charts effectively captured disparities among healthcare professions, illustrating that RTs consistently experience higher radiation doses compared to general doctors, nurses, and other medical professionals. These findings align with those of previous studies that identified RTs as a high-risk group due to their frequent exposure to diagnostic procedures [1,17]. This disparity was

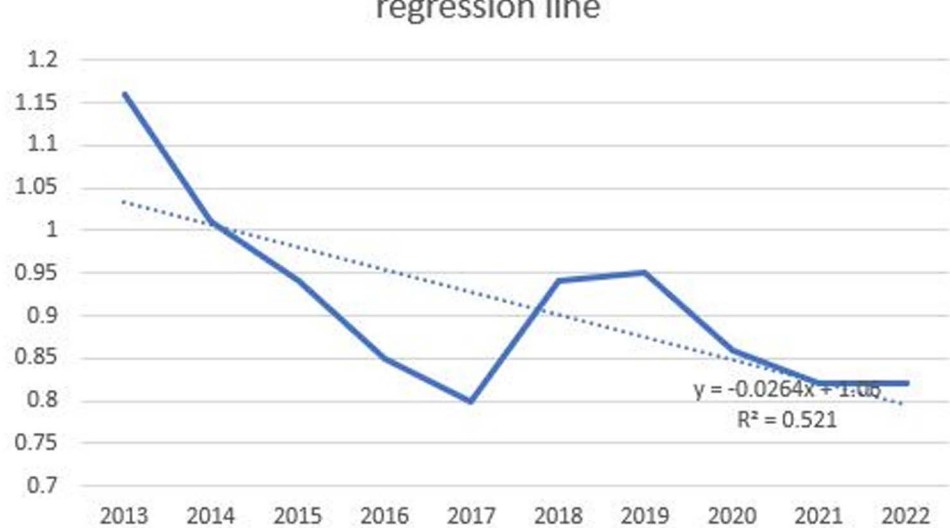

**Fig 11. Regression trend line illustrating the long-term trend in radiation exposure doses.** The figure demonstrates the effectiveness of safety management efforts over time.

particularly pronounced among younger and male RTs, indicating the need for targeted safety interventions and training programs to reduce occupational risks [4,6,8]. Box plots further emphasized the variations within the RT workforce, while sparklines revealed temporal trends, showing a steady decline in overall exposure levels since 2018 [4,9]. However, the average exposure dose for RTs remains significantly higher than that for healthcare professionals in other countries, such as the United Kingdom and Germany [4,16,18]. Geographic disparities, visualized through Chernoff faces and map charts, highlighted regions such as Jeonbuk with consistently higher exposure levels. These visualizations underscore the importance of regional safety measures and policy adjustments tailored to high-risk areas, as supported by recent occupational radiation guidelines [6,10,21].

The practical implications of these findings are significant. Policymakers and safety officers can use these visualizations to better allocate resources, prioritize interventions, and monitor the effectiveness of safety measures. For example, identifying high-risk demographic groups such as young male RTs enables more targeted outreach and education programs, while region-specific interventions can address localized risks. Furthermore, improved visualization techniques can empower radiological workers themselves by providing them with accessible and interpretable data, fostering greater awareness and engagement in safety practices [5,6].

Despite these contributions, this study has certain limitations that must be acknowledged. First, the reliance on pre-existing public datasets from the KDCA limits the granularity of the analysis. These datasets do not account for real-time exposure scenarios, a limitation also noted in previous public health studies that emphasize the importance of dynamic data systems [8,10,18]. Second, the visualization techniques employed in this study, although effective, are based on well-established methods such as radar charts and Chernoff faces. Innovative techniques such as augmented reality or interactive dashboards, which have been explored in related fields, could further enhance stakeholder engagement and usability [7,9,19]. While these tools are not within the scope of this study, their potential should be considered in future research.

Future research should address these limitations by integrating real-time data collection systems capable of capturing detailed exposure scenarios across various contexts. Such systems would enable more precise analyses of individual and

regional differences in radiation exposure levels. Additionally, advanced computational techniques, including multivariate regression models, clustering algorithms, and machine learning, could uncover deeper patterns and causal relationships within the data, as demonstrated in recent applications of machine learning in occupational health [8,15]. Furthermore, developing dynamic and interactive visualization platforms would enhance the accessibility of data for diverse stakeholders, from policymakers to radiological workers, fostering a more inclusive and informed approach to radiation safety management [6,9,20].

This study makes a substantial contribution to the field of occupational health by bridging the gap between raw radiation exposure data and actionable knowledge. The use of data visualization techniques not only facilitates better communication of complex datasets but also supports the development of safer working environments for radiological workers. These findings provide a foundation for evidence-based policymaking and the implementation of effective safety measures in radiation management, aligning with global recommendations on occupational radiation protection [5,6,8,21].

## Conclusions

This study demonstrates the transformative potential of data visualization techniques in improving the communication of radiation exposure data for radiological workers. By employing radar charts, box plots, sparklines, and Chernoff faces, the study effectively highlights disparities and trends, making complex datasets more accessible and actionable for stakeholders. The study findings emphasize the importance of integrating visualization methods into routine reporting frameworks to support evidence-based policymaking and improve safety management practices.

The results identified key risk factors for increased radiation exposure, including demographic disparities among younger and male RTs, as well as regional disparities in areas like Jeonbuk. These insights provide a foundation for targeted safety interventions and resource allocation. However, the study's reliance on static visualizations and pre-existing datasets presents limitations that should be addressed in future research. Real-time data collection systems, advanced computational techniques, and interactive visualization platforms offer promising directions for overcoming these challenges.

By bridging the gap between raw data and decision-making, this study contributes to the development of safer working environments and supports the implementation of evidence-based policies in radiation safety management. These methods have the potential to transform occupational health practices globally, enabling more informed and effective decision-making across healthcare systems.

## Acknowledgements

We would like to thank Editage (www.editage.co.kr) for English language editing.

## Author contributions

**Conceptualization:** Kyoungho Choi.

**Methodology:** Kyoungho Choi.

**Writing – original draft:** Kyoungho Choi.

**Writing – review & editing:** Kyoungho Choi.

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
