## [Decision Letter · Decision Letter 0]

11 Dec 2024

PONE-D-24-45344Importance of Data Visualization of Exposure Dose Statistics for Radiological Workers in Korean Healthcare FacilitiesPLOS ONE

Dear Dr. Choi,

Thank you for submitting your manuscript to PLOS ONE. After careful consideration, we feel that it has merit but does not fully meet PLOS ONE’s publication criteria as it currently stands. Therefore, we invite you to submit a revised version of the manuscript that addresses the points raised during the review process.

We look forward to receiving your revised manuscript.

Kind regards,

Mohamad Syazwan Mohd Sanusi

Academic Editor

PLOS ONE

Journal Requirements:

3. We note that Figure 10 in your submission contain map/satellite images which may be copyrighted. All PLOS content is published under the Creative Commons Attribution License (CC BY 4.0), which means that the manuscript, images, and Supporting Information files will be freely available online, and any third party is permitted to access, download, copy, distribute, and use these materials in any way, even commercially, with proper attribution. For these reasons, we cannot publish previously copyrighted maps or satellite images created using proprietary data, such as Google software (Google Maps, Street View, and Earth). For more information, see our copyright guidelines: http://journals.plos.org/plosone/s/licenses-and-copyright. We require you to either (a) present written permission from the copyright holder to publish these figures specifically under the CC BY 4.0 license, or (b) remove the figures from your submission:

a. You may seek permission from the original copyright holder of Figure 10 to publish the content specifically under the CC BY 4.0 license. We recommend that you contact the original copyright holder with the Content Permission Form (http://journals.plos.org/plosone/s/file?id=7c09/content-permission-form.pdf) and the following text: “I request permission for the open-access journal PLOS ONE to publish XXX under the Creative Commons Attribution License (CCAL) CC BY 4.0 (http://creativecommons.org/licenses/by/4.0/). Please be aware that this license allows unrestricted use and distribution, even commercially, by third parties. Please reply and provide explicit written permission to publish XXX under a CC BY license and complete the attached form.” Please upload the completed Content Permission Form or other proof of granted permissions as an "Other" file with your submission. In the figure caption of the copyrighted figure, please include the following text: “Reprinted from [ref] under a CC BY license, with permission from [name of publisher], original copyright [original copyright year].”

b. If you are unable to obtain permission from the original copyright holder to publish these figures under the CC BY 4.0 license or if the copyright holder’s requirements are incompatible with the CC BY 4.0 license, please either i) remove the figure or ii) supply a replacement figure that complies with the CC BY 4.0 license. Please check copyright information on all replacement figures and update the figure caption with source information. If applicable, please specify in the figure caption text when a figure is similar but not identical to the original image and is therefore for illustrative purposes only. The following resources for replacing copyrighted map figures may be helpful: USGS National Map Viewer (public domain): http://viewer.nationalmap.gov/viewer/ The Gateway to Astronaut Photography of Earth (public domain): http://eol.jsc.nasa.gov/sseop/clickmap/ Maps at the CIA (public domain): https://www.cia.gov/library/publications/the-world-factbook/index.html and https://www.cia.gov/library/publications/cia-maps-publications/index.html NASA Earth Observatory (public domain): http://earthobservatory.nasa.gov/ Landsat: http://landsat.visibleearth.nasa.gov/ USGS EROS (Earth Resources Observatory and Science (EROS) Center) (public domain): http://eros.usgs.gov/# Natural Earth (public domain): http://www.naturalearthdata.com/

Reviewers' comments:

Reviewer's Responses to Questions

**Comments to the Author**

1. Is the manuscript technically sound, and do the data support the conclusions?

Reviewer #1: Partly

Reviewer #2: Yes

2. Has the statistical analysis been performed appropriately and rigorously? 

Reviewer #1: N/A

Reviewer #2: N/A

3. Have the authors made all data underlying the findings in their manuscript fully available?

Reviewer #1: Yes

Reviewer #2: Yes

4. Is the manuscript presented in an intelligible fashion and written in standard English?

Reviewer #1: Yes

Reviewer #2: Yes

5. Review Comments to the Author

Reviewer #1: Given that this article does not focus on scientific testing or original research, but rather on improving data presentation through visualization, it might not be suitable for a high-impact investigational scientific journal that prioritizes novel research findings. Instead, this article might be more appropriate for journals focusing on public health, epidemiology, or data visualization—especially those that focus on improving communication or safety practices in industries like healthcare. However, this may yet be amenable to PLOS publication in some formats.

The article discusses methods for enhancing the communication of radiation exposure data for healthcare workers, particularly radiological technologists (RT), through data visualization techniques. It focuses on improving the 2022 Annual Report by the KDCA, which provides data on individual exposure doses but is primarily in tabular form, making it difficult for non-experts to understand.

Key points include:

Dose Comparisons: The report highlights that RT workers have significantly higher radiation exposure than other healthcare professions, such as dental doctors and general doctors. Visual tools like box plots and radar charts were used to show these differences, with radar charts revealing higher doses in younger and male RT workers.

Trends in Workforce and Exposure: The number of radiological workers has increased, with growth particularly among nurses and radiology residents. Sparkline charts and treemaps were recommended to show trends in workforce growth and regional variations in exposure doses.

Multivariate Data Visualization: Chernoff Faces were used to visualize the distribution of radiation doses across multiple regions and professional categories. These visual representations allowed for quick identification of high-risk areas, although there is some variability in how these faces are interpreted based on the data assigned to facial features.

Regulatory Impact on Exposure: Despite a general decline in average radiation exposure since 2018, RT workers still experience higher doses. Regression analysis showed a decreasing trend in RT exposure, indicating that safety measures have had a positive effect.

Visualization as a Solution: The article argues that data visualization methods like radar charts, sparklines, and Chernoff faces can make complex exposure data more accessible, especially for readers without a technical background. Visualization can highlight key trends, patterns, and comparisons, making it easier to convey important information than raw tables alone.

Limitations and Accessibility: While Excel and R programming were used for creating the visualizations, the study acknowledges that R may be challenging for some users, though it is a powerful tool for advanced visualizations.

Conclusion: The article concludes that data visualization is crucial for improving the delivery and understanding of exposure dose information. It recommends supplementing traditional tables with visual tools to make the data more intuitive and actionable, supporting better safety management for radiological workers.

Overall, the article advocates for a shift towards using more accessible, clear, and engaging visual tools to communicate radiation exposure data effectively, enhancing comprehension and aiding decision-making in safety management

Limitations:

Lack of a Scientific Hypothesis or Research Question: The paper does not test a specific scientific hypothesis or research question. Instead, it largely presents a collection of visualization methods applied to existing data from the KDCA's 2022 Annual Report. There is little direct experimentation or data analysis aimed at answering a novel scientific question about radiation exposure itself. The article is more about how to present data rather than generating new insights from the data itself.

Absence of Original Data Collection or Experimental Results: The study doesn't collect new data or provide new findings about the radiation exposure trends. It uses already available public datasets and focuses on presenting them in a more accessible format. While this is a useful exercise, it's more of an informational study or tool development rather than a groundbreaking piece of research.

Limited Statistical Testing: While the paper mentions some statistical tests (e.g., regression lines for trends), it doesn’t appear to use statistical methods to draw conclusions from the data. It could benefit from a more robust statistical analysis or hypothesis testing around a scientific question to examine the impact of different variables (e.g., profession, age, sex) on radiation exposure more rigorously.

Limited Novelty in Visualization Techniques: While the article does a good job of applying existing techniques (radar charts, sparklines, etc.) to the problem, these methods are not particularly novel. There is limited innovation in the choice of tools or in how these tools are applied to the dataset.

Reviewer #2: The manuscript was good interesting and good written.

Introduction was good written and include the aim of the study.

Materials and methods were good designed.

Results were good illustrated.

Discussion was good written.

6. PLOS authors have the option to publish the peer review history of their article (what does this mean?). If published, this will include your full peer review and any attached files.

Reviewer #1: No

Reviewer #2: No

---

## [Author Response · Author response to Decision Letter 0]

5 Jan 2025

The response letter addressing the comments from the reviewers and editor has been attached as a file.

---

## [Decision Letter · Decision Letter 1]

11 Feb 2025

PONE-D-24-45344R1Enhancing the Communication of Radiation Exposure Data for Radiological Workers Using Data Visualization TechniquesPLOS ONE

Dear Dr. Choi,

Thank you for submitting your manuscript to PLOS ONE. After careful consideration, we feel that it has merit but does not fully meet PLOS ONE’s publication criteria as it currently stands. Therefore, we invite you to submit a revised version of the manuscript that addresses the points raised during the review process.

We look forward to receiving your revised manuscript.

Kind regards,

Mohamad Syazwan Mohd Sanusi

Academic Editor

PLOS ONE

Journal Requirements:

Additional Editor Comments:

Dear Authors,

Please find my comments below for your reference:

Title: Please include the state or country where the proposed method is applied. If this method specifically targets the health sciences community in a particular region (e.g., Asia), please indicate this in both the abstract and introduction.

Images and Tables: Improve the quality of the images presented in this work. All tables must be revised; some need to be enlarged with clearer text and color.

Column Titles: Highlight some column titles to improve visibility.

Captions: Ensure all figures have appropriate captions.

Capitalization: Maintain consistent capitalization for place names, using either title case or sentence case as appropriate.

Units: Provide appropriate units for all figures and tables (e.g., year, age, crime rate).

Typo: Correct "Categoty" to "Category."

Spacing: Ensure "0.38 mSv" includes a space between the number and unit.

Line 81: Remove the quotation ("") symbol.

Figure 5 Caption: Does it need to specify that it is a U.S. state map? If so, please cite the reference as well.

Reviewers' comments:

Reviewer's Responses to Questions

**Comments to the Author**

1. If the authors have adequately addressed your comments raised in a previous round of review and you feel that this manuscript is now acceptable for publication, you may indicate that here to bypass the “Comments to the Author” section, enter your conflict of interest statement in the “Confidential to Editor” section, and submit your "Accept" recommendation.

Reviewer #1: (No Response)

Reviewer #2: All comments have been addressed

2. Is the manuscript technically sound, and do the data support the conclusions?

Reviewer #1: Yes

Reviewer #2: Yes

3. Has the statistical analysis been performed appropriately and rigorously? 

Reviewer #1: Yes

Reviewer #2: Yes

4. Have the authors made all data underlying the findings in their manuscript fully available?

Reviewer #1: Yes

Reviewer #2: Yes

5. Is the manuscript presented in an intelligible fashion and written in standard English?

Reviewer #1: Yes

Reviewer #2: Yes

6. Review Comments to the Author

Reviewer #1: Generalizability:

The study is focused on data from South Korea and may not necessarily be generalizable to other regions or countries. Differences in healthcare infrastructure, radiation safety regulations, and exposure practices could lead to different results in other settings. Cross-country comparisons should account for these contextual differences when making conclusions about radiation safety on a global scale.

Visualization Over-reliance:

While the article advocates for using visualization techniques to simplify complex data, there is a risk of over-relying on visualizations without ensuring that they provide enough depth for decision-making. Visualization methods like radar charts or Chernoff faces are useful for highlighting patterns but may not provide sufficient statistical rigor or details required for comprehensive scientific analysis or policy development.

Ethical and Privacy Concerns:

Although not explicitly addressed in the article, the study uses individual exposure data from healthcare workers. It is important to ensure that privacy and ethical considerations are taken into account when handling and presenting sensitive data. The article does not mention any safeguards for ensuring that the data is anonymized or that privacy concerns are adequately addressed.

ChatGPT can make mistakes. Check important info.

Reviewer #2: The revised version of the manuscript was good written and all comments of reviewers were addressed.

7. PLOS authors have the option to publish the peer review history of their article (what does this mean?). If published, this will include your full peer review and any attached files.

Reviewer #1: No

Reviewer #2: No

---

## [Author Response · Author response to Decision Letter 1]

6 Mar 2025

The responses to the editor and reviewers have been attached as a file.

---

## [Decision Letter · Decision Letter 2]

2 Apr 2025

Enhancing the Communication of Radiation Exposure Data for Radiological Workers in Kortea Using Data Visualization Techniques

PONE-D-24-45344R2

Dear Dr. Choi,

We’re pleased to inform you that your manuscript has been judged scientifically suitable for publication and will be formally accepted for publication once it meets all outstanding technical requirements.

Kind regards,

Mohamad Syazwan Mohd Sanusi

Academic Editor

PLOS ONE

Additional Editor Comments (optional):

Reviewers' comments:

Reviewer's Responses to Questions

**Comments to the Author**

1. If the authors have adequately addressed your comments raised in a previous round of review and you feel that this manuscript is now acceptable for publication, you may indicate that here to bypass the “Comments to the Author” section, enter your conflict of interest statement in the “Confidential to Editor” section, and submit your "Accept" recommendation.

Reviewer #1: All comments have been addressed

Reviewer #2: All comments have been addressed

2. Is the manuscript technically sound, and do the data support the conclusions?

Reviewer #1: Yes

Reviewer #2: Yes

3. Has the statistical analysis been performed appropriately and rigorously? 

Reviewer #1: Yes

Reviewer #2: Yes

4. Have the authors made all data underlying the findings in their manuscript fully available?

Reviewer #1: Yes

Reviewer #2: Yes

5. Is the manuscript presented in an intelligible fashion and written in standard English?

Reviewer #1: Yes

Reviewer #2: Yes

6. Review Comments to the Author

Reviewer #1: Comments were addressed and placed in the document with tracked changes. Thanks for the opportunity to review.

Reviewer #2: The revised manuscript was good written and discussed.

All comments of reviewers were addressed in the manuscript.

7. PLOS authors have the option to publish the peer review history of their article (what does this mean?). If published, this will include your full peer review and any attached files.

Reviewer #1: No

Reviewer #2: No

---

## [Editor Report · Acceptance letter]

PONE-D-24-45344R2

PLOS ONE

Dear Dr. Choi,

I'm pleased to inform you that your manuscript has been deemed suitable for publication in PLOS ONE. Congratulations! Your manuscript is now being handed over to our production team.

Kind regards,

on behalf of

Dr. Mohamad Syazwan Mohd Sanusi

Academic Editor

PLOS ONE